# OBINTER: A Holistic Approach to Catalyse the Self-Management of Chronic Obesity

**DOI:** 10.3390/s20185060

**Published:** 2020-09-06

**Authors:** Roberto Álvarez, Jordi Torres, Garazi Artola, Gorka Epelde, Sara Arranz, Gerard Marrugat

**Affiliations:** 1Vicomtech Foundation, Basque Research and Technology Alliance (BRTA), Mikeletegi 57, 20009 Donostia-San Sebastián, Spain; jtorres@vicomtech.org (J.T.); gartola@vicomtech.org (G.A.); 2Biodonostia Health Research Institute, eHealth Group, 20014 San Sebastián, Spain; 3AZTI, Food Research, Basque Research and Technology Alliance (BRTA), Parque Tecnológico de Bizkaia, Astondo Bidea, Edificio 609, 48160 Derio, Spain; sarranz@azti.es (S.A.); gmarrugat@azti.es (G.M.)

**Keywords:** self-management, obesity, personalized recommendations, adherence, patient-reported outcomes

## Abstract

Obesity is a preventable chronic condition that, in 2016, affected more than 1.9 billion people globally. Several factors have been identified that have a positive impact on long-term weight loss programs such as personalized recommendations, adherence strategies, weight and diet follow-up or physical activity tracking. Recently, various applications have been developed which help patients to self-manage their condition. These apps implement either one or some of these identified factors; however, there is not a single application that combines all of them following a holistic approach. In this context, we developed the OBINTER platform, which assists patients during the weight loss process by targeting user engagement during the longer term. The solution includes a mobile application which allows users to fill out dietetic questionnaires, receive dietetic and nutraceutical plans, track the evolution of their weight and adherence to the diet, as well as track their physical activity via a wearable device. Furthermore, an adherence strategy has been developed as a tool to foster the app usage during the whole weight loss process. In this paper, we present how the OBINTER approach gathers all of these features as well as the positive results of a usability testing study performed to assess the performance and usability of the OBINTER platform.

## 1. Introduction

Obesity is a chronic disease with a multifactorial aetiology including genetics, environmental, metabolism, lifestyle, and behavioural components [1]. This condition is closely related to several other chronic diseases, including heart disease, hypertension, type 2 diabetes, sleep apnoea, certain cancers, joint diseases, and more [2]. The worldwide prevalence of obesity nearly doubled between 1980 and 2008. In 2008, the World Health Organization (WHO) reported through its Regional Office for Europe that over 50% of both men and women in the WHO European Region were overweight, and roughly 23% of women and 20% of men were obese [3]. Additionally, in the current COVID-19 pandemic, it is important to highlight the severity of the COVID-19 cases identified on people with obesity [4,5,6] and closely related type 2 diabetes [7].

In view of this, the use of m-health app and wearable devices for improving health care processes and outcomes provide low-cost, effective care and healthy lifestyle promotion for patients with obesity. However, there is a lack of expert involvement together with a lack of adherence to medical evidence in most of those m-health apps [8]. This also applies for apps used for overweight and obesity treatment. In addition, although weight loss interventions typically have consisted of combined strategies including diet, exercise, and behaviour therapy [9], current apps offer a limited combination of them.

In this paper, we focus on implementing a holistic approach named OBINTER which leverages, in a unified way, multiple techniques that have proved significant outcomes individually or in limited combination. The set of techniques addressed are (1) patient-reported outcomes (PROs); (2) personalized dietetic and nutraceutical recommendations based not only on participants’ PRO, but also considering their molecular data (cell membrane lipidomic and biochemical analysis); d3) Diet adherence control; (4) physical activity tracking; (5) weight evolution control; and (6) adherence strategy.

Moreover, the majority of the mobile applications solutions are designed to be controlled by health professionals; however, in this paper, we propose a digital technology promoting self-monitoring of physical activity and diet adapted to different individual profiles, taking into account nutritional, metabolic and behavioural needs of participants to offer personalized feedback to the user.

Finally, this holistic approach has undergone a user evaluation protocol carefully defined, run and assessed in order to evaluate both the OBINTER approach as well as the developed solution’s usability.

This paper is structured as follows: Section 2 describes the state of the art related to our topic and underlines the main contributions our approach provides with respect to them; in Section 3, the general OBINTER platform and all the currently supported strategies are covered. Additionally, the user evaluation protocol is presented; Section 4 presents the results of the usability study; Section 5 discusses the results in detail; Section 6 presents the conclusions and the future work inferred mainly from them.

## 2. Related Work

In this obesity global epidemic and involved risks context, different studies have been conducted to assess the efficacy of mobile applications used to provide health advice for weight loss. Although some of this research did not reveal significant differences in participants’ weight change compared to usual care [10,11], integrating the use of the app with health system resources (dietitian-delivered counselling calls, clinician counselling informed by app-generated recommendations) [12] or with social support [13] has proven to lead to clinically meaningful weight-loss outcomes. However, as shown in a study on the current weight management app market, apps lack professional content expertise as well as in evidence-based online approaches [14].

Several studies have concluded that the use of mobile apps for weight loss plays an important role in the adherence to dietary self-monitoring [15,16]. For example, Marcos et al. [17] have studied the impact of using a digital platform to increase adherence to the Mediterranean diet and physical activity (PA) in obese patients, demonstrating a significant difference between the weight reduction values among the participants in the clinical trial (greater weight loss for intervention patient group).

In recent years, health-related mobile apps have started using push notifications to improve their efficacy by more effectively engaging with users and consequently obtaining better results. As an example, the efficacy of push notifications was evaluated in a clinical trial aimed at improving the body composition of obese adult women after undergoing a dietary procedure [18]. They analysed the evolution of body composition based on push notifications and prescribed PA, proving the effectiveness of the proposed program.

The advent of wearable technologies, which encourages users to engage in PA and provides useful information for personalized care, has also positively influenced the prevention of overweight and obesity. Different studies have been conducted in this field [19], some of which demonstrate the effectiveness of wearables in weight control [20].

Although PA monitoring can be accurately measured with digital devices, food intake tracking through food frequency questionnaires is deemed insufficient in providing users with precise information on diet strategy without considering their metabolism [21].

As shown, current weight loss applications propose different strategies to improve the efficacy in the management of users’ weight, showing promising results. Nevertheless, they apply these strategies individually or they use a limited combination of them, which is not in accordance with the inherent complexity that obesity chronic condition has. According to [22], obesity has to be accepted as a complex problem and corresponding solutions have to address the multiple interconnections and elements that contribute to obesity if they want to successfully manage it.

Table 1 summarizes the functionality of some of the aforementioned applications compared with the dimensions present on the OBINTER App. Based on the review of Pagoto et al. [23], we have additionally included some of the most popular commercial apps for weight loss, calorie counting, etc. In particular, those whose contents are the most evidence based according to the author’s scale (5 out of 5): Lose It! MyNetDiary and MyFitnessPal.

## 3. Materials and Methods

The main objective OBINTER approach pursues is to provide final users with a robust tool capable of guaranteeing an efficient self-monitoring process of the obesity chronic condition. This objective is articulated by the aggregation of a set of unitary methods that rely on a platform that has been designed strongly considering its sustainability, scalability and maintainability. The platform as well as all of the combined methods are described within the rest of this section.

### 3.1. Platform

The general system architecture of the OBINTER platform can be described from three different points of view that as a whole reflect the foundations of the global approach. These perspectives are the physical view, the development view, and finally the logical view which partially follows the 4 + 1 architecture model [24].

i.Physical View

The OBINTER platform is physically composed of (1) a hybrid frontend implemented as a smartphone app, (2) a backend in charge of managing the business logic, including the authentication process, and managing the persistence capabilities by combining an SQL database that hosts all the structured data we have internally designed and a NoSQL database to store all the information whose format is not under our control (e.g., information coming directly from the wristband), and (3) an optional wristband tracker. In addition to these three elements, we should also mention two more physical third-party components our system relies on: (1) Firebase backend API to support the notification system explained in Section 3.7 and (2) Fitbit backend API to gather the data generated by the wristband (detailed in Section 3.5).

ii.Development View

From the development point of view, the OBINTER platform follows a hexagonal architecture [25] (also known as ports and adapters architecture) as the main methodology for its software design. As depicted in Figure 1, business logic constitutes the core of the platform and all the surrounding components are developed following an abstraction pattern that presents them as ports following a specific API that can be provided with different adapters. This way, the resultant system is built upon a set of loosely coupled components that might even be easily interchangeable by others following the same API. By adopting this architecture, the OBINTER approach intends to facilitate system maintenance as well as the platform testing process. This second point may become a critical factor in closing the existing gap between experimental developments and production-ready implementations.

iii.Logical View

The aim of the logical view is centred on describing the functionality that the system provides to end-users. From this point of view, the set of functionalities the OBINTER platform provides are (i) gathering patient-reported outcomes (Section 3.2 describes this functionality), (ii) delivering dietetic and nutraceutical recommendations (detailed in Section 3.3), (iii) following-up on critical features such as users’ diet adherence (Section 3.4) and weight evolution (Section 3.6), (iv) delivering an adherent strategy to ensure users’ compromise (this strategy is outlined in greater depth in Section 3.7), and optionally, (v) visualizing physical activity parameters (detailed in Section 3.5).

To make the application available to both Android and iOS users, a cross-platform development was used. More specifically, the Ionic framework [26] was chosen to develop the application due to its hybrid nature. The resulting application can then be uploaded to the Google Play Store^®^ as well as Apple Store^®^.

Figure 2 depicts the resulting high-level overview after combining the three previously described views, which also details the data-flow interactions.

### 3.2. Patient-Reported Outcomes (PROs)

A Patient-Reported Outcome (PRO) refers to information about the health status of a patient provided directly by himself, without having previously been interpreted by a doctor or a specialist [27]. PRO reports are acquiring significant relevance as clinical assistance models are gradually adopting a patient-centred philosophy. As reported by the European Patient’s Academy [28], there is an increasing awareness that treatments should not only be clinically effective and economically efficient but should also be acceptable and indeed desirable for patients. These are factors that can only be gathered by patient-based evidence that includes measures of well-being.

As stated in a white paper by the NIH Collaboratory [29], as the use of PROs in both research and routine patient care has increased, so has the use of mobile technologies (such as smartphones or wearables). These technologies, known collectively as “mobile health” are utilized in OBINTER as means by which information directly derived from patients can be incorporated into its Information System with the aim of:Assessing both the initial and final general patient background and current status using as PRO instrument a questionnaire (see Figure 3) designed by OBINTER’s research team that encompasses a multidisciplinary group of experts with relevant expertise in Digital Health, Nutrition, Endocrinology and Clinical Practice. The user responds to a life-habits questionnaire at the beginning and end of the study that is based on published, validated questionnaires in order to obtain information from the adult population regarding physical activity (IPAQ) [30] and Food Frequency intake [31]. The answers obtained in conjunction with red blood cell membrane lipidomic profiles [32] of each user are applied to generate personalized dietetic and nutraceutical recommendations. This process is detailed in Section 3.3.Gathering physical measurements that occur during the patient’s daily life using a wearable as PRO instrument that will accompany the user throughout the study. The specific measurements that are gathered as well as the integration mechanisms are further described in Section 3.5.Following up on treatment adherence and effectiveness using as PRO instrument questionnaires delivered directly to the users. These two aspects are described in greater detail in Section 3.4 and Section 3.6.

### 3.3. Personalized Dietetic and Nutraceutical Recommendations

When users have answered the initial questionnaire, a recommendation system will compute their responses together with their lipidomic profile, generating as a result personalized recommendation. Users’ fatty acid composition of red blood cell membrane is obtained from a blood sample extracted at the beginning of the intervention. The cell membrane lipidomic profile accurately describes the subject’s metabolic status and represents an overview of the dietary habits the individual followed over the prior four months. In addition, the cell membrane lipidomic profile is capable of characterizing metabolomic disorders as well as nutrient deficiencies in a diet. Through analysis of the bio-indicators, it is possible to design a more personalized and effective obesity treatment. 

Personalized recommendations will be received in the mobile application and shown in a simple and schematic way to improve their understanding, as illustrated in Figure 4. It has been proven that digital tools allow to gather information about behaviour patterns in different population groups [33]; additionally, capturing their lifestyle and communicating with the user through an external device, the mobile phone, makes the intervention less invasive. Recommendations include dietetic guidelines and nutraceutical supplementation if needed. The nutritional advice provided is based on the Mediterranean diet [34], empowering and restricting some food according to macronutrients (fats, carbohydrates, and proteins) and micronutrients (omega-3, vitamins, and minerals) along with the individual’s requirements. The recommendations given intend to influence the user’s eating behaviour and increase adherence to a well-balanced diet properly adapted to their geographical area as well as the user’s needs.

### 3.4. Diet Adherence Control

After receiving the dietetic recommendation, users will be encouraged to respond, at least weekly, to a 14-item questionnaire based on a previously validated questionnaire (Mediterranean Diet Adherence Screener, MEDAS) [35] in order to quantify their adherence to the diet. The responses are summarized in a score whose evolution is depicted to the users. Both the questionnaire as well as an example of a user’s evolution of responses is conveniently illustrated below in Figure 5.

In order to engage users in the active use of the application, and therefore to achieve higher adherences to the diet, an adherence system among members has been implemented that takes into consideration their diet adherence scores. This strategy is more broadly addressed in Section 3.7. The expected results are, on the one hand, an increase of the dietetical recommendations adherence scores and as a consequence the improvement of users’ health, and on the other hand an improvement of the evaluation results in digital tools usability and acceptance.

### 3.5. Physical Activity Tracking

The adoption of wearable devices is allowing Digital Health solutions to have an enormously valuable complementary source of data. Therefore, the integration capability with wearables is one of the main functionalities the OBINTER platform is able to provide. As explained in Section 3.1, the platform provides this feature as a port that currently has only one adapter implemented but could easily be extended at any time. This adaptation has been carried out upon the wristband tracker Fitbit^®^ Inspire HR [36], which was selected due to (1) its ability to track all of the parameters identified by OBINTER’s researchers as relevant for the obesity condition: heart rate, calories burned, daily steps, active minutes, stationary time, sleep quality and floors climbed; (2) it provides a competitive API to ease its integration in third-party apps; and, finally, (3) its cost/benefit ratio is rather good.

The Wearable Module is in charge of interacting with the Fitbit API. These interactions require a Fitbit Access Token that is obtained when the user gives consent in the OBINTER application to access its data. The authentication process is detailed in Figure 6.

Obtaining user’s consent from the application is possible as the OBINTER platform is registered on the Fitbit website. This way, with the user’s consented token, calls to the API can be made directly from the Wearable Module. The data obtained from the calls is comprised of all the above-mentioned parameters. In addition, for each of these fields, two aggregated charts are also depicted to allow users in order to provide better insights into their physical data: (1) the weekly averages since the beginning of the OBINTER platform use, and (2) the daily aggregated value throughout the last week. An example of these charts can be shown on Figure 7.

To allow users to have complete control over their data, they have the option to revoke their consent at any time. The stored tokens are revoked and deleted from the database, and the authorization process will be repeated if the user decides to give their consent again in the future.

### 3.6. Weight Evolution Control

Together with the PA tracking, weight evolution monitoring is another key factor our platform manages. Users can introduce their weight into the OBINTER app as many times as they want, but they are encouraged to do so at least once weekly. In addition, they are advised to do so always at the same time of the day and to use the same scale. In this weight introduction process, the system asks users to specify if they think they have gained, lost, or maintained their previous weight prior to weighing themselves, in order to not only gather their specific weight, but also to obtain users’ perception of their personal evolution. Additionally, the app will show a weight evolution graph, where users can visualize their last seven days’ weight information. In addition, the positive or negative value of the difference between the first and last introduced weights is displayed in the weight control Table This value allows users to have a quick view of their progress, as can be seen in Figure 8.

### 3.7. Adherence Strategy

Patient compliance focuses on describing the degree to which a patient correctly follows health-related prescriptions. These dispositions have been historically related to drug compliance, but nowadays also applies, in an ever-increasing way, to other aspects such as smartphone app use, self-care, self-directed exercise or therapy sessions.

Adherence is a two-sided coin where optimal compliance conforms an obvious cornerstone on success in the management of people’s condition(s). At the same time, however, adherence it is also a major obstacle in the effective delivery of health care in those scenarios where compliance is not achieved. According to a 2003 report by the World Health Organization [37], adherence to long-term chronic disease therapies is low (with an average of 50% in developed countries, while developing countries show lower adherence rates), and it is unquestionable that many patients struggle to correctly follow prescribed therapy or advised recommendations.

Historically, adherence has been a concept bound exclusively to medication. In the relevant literature [38], we can find contrasted methods for both measuring the adherence and improving it. The OBINTER approach has followed those medication-related baselines and adapted them when possible in order to concrete implementations in a wider scenario that involves Digital Health solutions. According to this course of action, we have designed and implemented the following adherence strategies:Methods of Measuring Adherence:
∘Patient questionnaires or patient self-reports are indirect methods that allow users to inform themselves about their status and progresses. The main disadvantage here, as it is in the traditional approach, is that results can be easily distorted by the patient. To mitigate this problem, the OBINTER platform checks user inputs before adding them to Information System trying to prevent the insertion of low-quality data by means of applying a different set of statistical methods or reference values, depending on the nature of the questionnaire, that could spot unusual values. For example, according to the NHS [39], a safe rate for losing weight is 0.5kg to 1kg each week, so values above/below them will be internally considered as unusual and suggested to be confirmed twice by the user.∘Measurement of physical/physiologic markers (as an indirect method of measuring adherence). The integration of the wearable device becomes a relevant source of information to discover not only if users synchronize their data regularly, which would mean a regular interaction with the application, but also to explore and follow up on those parameters directly related to the obesity condition that might be a beacon for foreseeing low levels of adherence to the dietetic and nutraceutical recommendations.Methods for Improving Adherence:
∘Identify poor adherence: closely related to the methods for measuring adherence, but also complemented with other measurements such as login attempts or low personal scores in the virtual race (explained in next bullet). Since all the interactions carried out by the users within the system are tracked, OBINTER has mechanisms to detect poor adherence which is the first step to solving this issue.∘Emphasize the value of the regimen and the effect of adherence: OBINTER’s participants are randomly assigned to a team with which they will run a virtual race “against” obesity (Figure 9). Users are informed that all their interactions and positive scores will contribute with additional virtual meters for their corresponding teams. This way, users are invited to relate compliance with a double positive purpose, therefore emphasizing the value of a good adherence. In addition, since users run in teams, we avoid that they view themselves as solely responsible for the task and simultaneously promote teamwork which has been proven as more effective than individual responsibility in multiple scenarios [40].∘Provide simple, clear instructions and simplify the regimen as much as possible: best practices for designing an interface [41] have been adopted during OBINTER’s implementation process. The interface has been kept as simple as possible, always using common and consistent UI elements. Careful placement of items was done to help draw attention to the most important pieces of information. Internationalization issues were also considered, since the application is going to be tested in a two-official-languages country. In addition, nutraceutical and dietetic recommendations are depicted in a straightforward manner aimed at avoiding problematic misunderstandings.∘Reinforce desirable behaviour and results when appropriate: to pursue this reinforcement, OBINTER’s platform makes use of its own notification system designed to remind the users the correct timeline physicians have designed for them to follow. Push notifications are delivered just before a scheduled relevant task. In addition, the timeline is always available the user’s consultations in the main section of the application. A second implementation has been carried out to reinforce desirable behaviour. This second implementation is a challenge delivery system that encourages users to achieve specific actions aimed to both advance in their obesity management and contribute to the teamwork, since the achievement of these challenges yields additional virtual meters for their teams (Figure 9).

### 3.8. User Evaluation Method

To evaluate the solution reported in this paper, a usability testing approach was adopted. A usability test focuses on evaluating how well real users are able to accomplish representative tasks of the software, while an experimenter collects the problems and errors made during the experiment. This methodology has already been broadly verified for medical software [42,43]. Considering the previous points, we have adopted this methodology for this study. Considering the holistic approach for the self-management of obesity proposed in this paper, representative tasks have been selected to evaluate the different dimensions covered by the OBINTER approach.

#### 3.8.1. Study Design

A detailed test protocol was developed, inspired by the protocol detailed in [44]. The protocol is depicted graphically in Figure 10. The evaluation material was developed in Spanish and the Spanish version of the application was tested with all users (the application is available in both Basque and Spanish languages) to avoid translation biases on the experiment (all assessed Basque speakers are advanced Spanish speakers).

As a starting point and prior to the usability testing, all participants were provided with an information sheet on the goals and means of the evaluation and signed a voluntary consent form allowing for screen and audio recordings (to capture user interactions and subjects verbalization while performing the tests’ tasks, recordings were securely kept and deleted as soon as they were processed). Participant information sheet and consent form are available as Appendix A, in both English and Spanish languages (Spanish version was used in the evaluations, while English version is provided for the understanding of non-Spanish speakers). Next, before starting the test, each participant was asked to complete a pre-questionnaire aimed to later analyse the different profiles of the participants. This questionnaire consists of nine questions that try to collect the participants’ gender, age, occupation, primary experience, first language, computer skills, and familiarity with diet/wellness applications.

The familiarity of the participant has been consulted by means of a question that is answered as a five-point Likert psychometric scale (one means “low” familiarity while five means “high” familiarity).

Following, each subject was instructed to ‘think-aloud’ while trying to achieve the defined tasks for the usability testing. At this point, subjects were told to briefly practice thinking aloud while doing an initial exploration of the App. The Think-Aloud Protocol (TAP) is a method that helps the experiment observer to understand the cognitive processes that may have given rise to usability problems or errors during the execution of the tasks requested to the participant [45].

Additionally, to capture and analyse tasks’ difficulty expectation and perception, before and after the real task execution experience, each participant was assessed following the Single Ease Question (SEQ) method [46].

Once participants were instructed and assessed, the experimenter asked participants to try to accomplish the defined set of representative tasks.

After each participant carried out all tasks, they then completed post-test questionnaires. The post-test questionnaire was composed of an extended System Usability Scale (SUS) and some questions targeting proposed holistic approach’s validity and acceptance. SUS instrument [47] was used for assessing the perception of the ease of using a system. The SUS instrument was extended with two additional questions, one to measure the Adjective Rating Scale (to summarize and better describe the SUS numeric score in an absolute judgment of usability through adjectives) [48] and the second one to capture the Net Promoter Score (NPS—to assess how likely a user would recommend this app to a friend or a colleague) [49].

Finally, three additional statements were presented to participants to assess OBINTER approach’s acceptance, using the same scoring as SUS (1 = ’Strongly Disagree’ and 5 = ’Strongly Agree’). The three statement were (translated from Spanish to English for the article):“I find the team competition system and challenge system motivating”;“The application can help me manage the different dimensions related to my overweight, in an orderly and easy way”;“I think that the proposed digital approach can help me manage my overweight and improve my well-being”.

#### 3.8.2. Representative Tasks

As it has been introduced in the study design section, after completing the pre-questionnaire, the moderator asked participants to perform the representative tasks defined for the study. These characteristic tasks, listed in Table 2, were agreed between the developers and the potential users of the application with the objective of evaluating the system’s essential features. For the wording of the tasks it is important not to give clues that might help in the accomplishment of the task. The six tasks defined for participants evaluation represent the different dimensions of the OBINTER approach.

For the usability tests, physical activity trackers were not configured, but the developed visualization interfaces were tested showing data from a physical activity tracker recorded by one of the developers.

#### 3.8.3. Hardware and Software Requirements

Due to the COVID-19 pandemic context initially planned face-to-face usability testing had to be redefined to be carried out remotely. The defined setting required a secure online deployment of the platform, a PC on the moderator’s side and both a PC and a smartphone on the tester’s side. PC-based interaction through a virtual meeting service was used to support the initial steps of introducing the goal of evaluation, soliciting audio and screen-capturing consent as well as gathering pre- and post-test questionnaires. The commercial Lookback tool [50] was used to remotely accomplish usability testing for the OBINTER App so the tester is required to install this small application in addition to the targeted OBINTER App evaluations (available for both iOS and Android) on the smartphone, which allows the users to carry out a virtual meeting while simultaneously capturing and recording the tester’s screen and audio.

#### 3.8.4. Script-Driven Study

The usability study has been supported by a common script prepared to approach the different participants in a systematic way.

As an example, when using the think-aloud protocol it is important to remind users to verbalize what they are thinking while performing tasks. In this sense, the script was prepared in such a way that at the beginning of each session, participants were reminded to speak aloud, indicating their objectives and their mental approach to the steps to complete the task.

The script includes numbered steps for administrative matters (e.g., informed consent, questionnaire delivery), task descriptions, and key actions for the evaluation (e.g., when to start recording the screen and audio).

The evaluation was carried out remotely in virtual meetings. During these virtual meetings, the script was presented to the user, recommending that they keep it in the foreground. Although each task was read to the participant before its execution, it was deemed important that the participant could re-read the content of the same, especially if the task was complex or required the execution of numerous steps.

## 4. Results

### 4.1. User Evaluation Results

Experimenters took notes and recorded both the screen as well as the audio of the evaluation session. The analysis of this content was carried out offline once the study was completed. Analysis results were saved in a shared document, and individual and aggregate analysis of the results was performed. As a result of this analysis, a compilation of demographic data, task completion rates and times, SEQ scores per task, and scores from the usability questionnaires was obtained. In addition, a compilation and analysis of the qualitative comments and ideas for improvement (collected during the evaluation) was carried out. Below, we summarize the outcomes and findings identified during the analysis of the evaluation results.

### 4.2. User Demographics

Table 3 summarizes the demographic data obtained through the pre-questionnaires prior to the evaluation.

The number of study participants (*n* = 10) is considered sufficient, taking into account the recommendation by usability experts of five participants to maximize cost/benefit of the evaluations [51]. According to these experts, 80% of the problems of a software application can be identified by only five participants [52]. No incentives were offered to study participants to avoid bias (e.g., if users are paid to participate in the evaluation, they may feel forced to give a positive evaluation of the software).

### 4.3. Task Completion and Times

Regarding task completion, all but one user completed every task, despite the help provided by the moderator. This user failed to complete Task 5, related to team assignment and race position of the team. Together with Task 1 (initial questionnaire on patients’ background—this task is longer than others in an order of magnitude), Task 5 is the task most frequently required the moderator’s assistance for completion. See Figure 11 for details regarding accomplishment per task. Task 1, two help aids were mostly motivated by questionnaires or sections descriptions not being clear enough or being strangely located (e.g., Nutraceutical section naming was not linked with the nutritional supplement concept requested, or vegetable drinks were sought together with different milk types consumption questionnaires), Task 3 (task on filling out diet information and weight follow-up) help aid was mostly related to the application interaction allowing the user to continue without filling in a response to a question, or not allowing the user to set a value to 0 (similar issues happened in some parts of the Task 1 evaluation). Task 5 help required was motivated by the scoring approach not being stated clearly enough (or prepared for diagonal readers or users quickly screening through the application). For Task 6 (on finding the proposed challenge and accepting it), some user users required help locating the challenges section. Specific issues are explored in more detail in the Qualitative Analysis section (below).

Concerning task duration analysis, Tasks 1 and 2 duration analysis has been split in two. Task 1 is larger by an order of magnitude, therefore has been analysed and plotted separately (See Figure 12 and Figure 13 for Task 1 participant-specific and overall results, and Figure 14 and Figure 15 for Tasks 2–6 participant-specific and overall results).

Task 1 mean duration is 1108 ± 368 s. Looking at Figure 12 and Figure 13, we can see that there is a large variance between those performing the task in 1000 s or less, and those performing this task in over 1300 s. Looking at the demographics data, the former group has reported a substantial higher computer and Internet usage per week. Previous experience with diet/wellbeing apps did not show any effect on the task duration or accomplishment.

In regard to the duration of Tasks 2–6 (see Figure 14 and Figure 15), Tasks 2, 3 and 5 follow a similar per-user participant pattern compared to Task 1. Tasks 3, 5 and 6 are the ones that show most variability, which is explained by a somehow lengthy (and similar interaction to Task 1) diet follow-up inputting in Task 3, and Task 5 being an exploratory task and requiring interpretation. Task 6 variability is caused by those users having opened (or noticed) the challenges section as part of a previous task, which also led to have a completely different per participant duration pattern compared to other Tasks (some users even get a better completion time compared to the expert user). For Task 4, there is a value concentration as shown by the different quartiles, which makes not very distant results to be marked as outliers. All participants completed Task 4 without assistance, and the time difference is related to the time required to discover the interaction to access the history plots.

### 4.4. SEQ Results Summary

The analysis of Single Ease Question (SEQ) results points out that users found platform use slightly easier than initially expected (see mean values improvement from SEQ_0 to SEQ for all tasks in Figure 16). Additionally, all post-task assessments have a mean score above 4, which should be considered as good score or perceived as easy to use. The Task 1 SEQ_0 outlier can be understood in the context of a non-experienced user scoring difficulty before the first interaction with the tool. The Task 2 and 4 outliers are plotted as outliers, but a quick view shows that it is motivated by a score of 4 when all other scores have been of 5. The Task 5 outlier was scored by a participant which deemed the information for team ranking unclear and required assistance to understanding the scoring logic behind this task. Task 5 has the lowest scoring (mean 4.1 ± 0.9); however, this is not a bad score, as shown in the participants’ request to better describe the gaming and scoring strategy implemented.

### 4.5. Extended SUS & Approach Acceptance Summary

The mean SUS score has been of 88.5 ± 8.4, which is a good result considering the reported average SUS score is 68/100. The raw SUS score is a single number representing a composite measure of the overall usability of the system being studied. Although SUS was originally designed to assess perceived usability as a single attribute, Lewis and Sauro [53] found that there are actually two factors in SUS. Eight of the questions reflect a usability factor and two reflect a learnability factor [54]. Figure 17 and Figure 18 show results per participant of the different SUS dimensions (Overall Score, Usability Dimension, Learnability Dimension) and the distribution of the aggregated results considering all tested tasks.

Results in Figure 17 and Figure 18 show that all three dimensions have a mean score around 90 with a limited distribution of scores (Overall SUS 88.5 ± 8.4, Usability dimension 87.8 ± 9, and Learnability dimension 91.2 ± 19.5). A specific outlier has been identified for Learnability dimension and for participant P_GE1, which after revising scores at different usability evaluation sections, it is concluded to be a misinterpretation of the SUS question number 4 on needing the help of a technical expert (the participant completed all tasks and scored them as easy in the post task SEQ assessment). Additionally, all of the other participants score learnability slightly higher than overall and specific usability dimensions.

As it can be seen, the SUS and usability values follow a similar pattern, while the learnability scores track higher in almost all the results. This is a clear indicator of the relationship between the SUS and the other two values introduced above. In addition, the learnability has had a higher average score with a 91.2, demonstrating that participants were able to use the application without any prior explanation.

With respect to the additional last two questions extending the SUS questionnaire, the adjective average obtained values of 5 and 6 (Good and Excellent), which resulted in an average of 5.7 ± 0.5. Thus, the participants have concluded that the overall user friendliness of our application is considered nearly excellent. The promoter rating scores (addressing the likeliness of participants recommending this app to others) were also high, with values between 7 and 9, obtaining a final NPS of 42% or +42. According to Reichheld [55], the highest performing organizations are situated between +50 and +80, thus our application does not achieve a bad score, considering that it has achieved five promoters (scoring 9 or 10), five neutral users (scoring 7 and 8) and no detractors (scoring 0 to 6).

These results are shown in Figure 19 and Figure 20.

Regarding approach specific statements evaluation, the first statement received a mean score of 3.5 ± 1.5. The second statement received a mean score of 4.6 ± 0.5. The third statement received a mean score of 4.7 ± 0.5. The second and third statement were similarly scored as they were formulated in the same way, even though the second question targeted the tool’s usability to deal with the specific issue and the third addressed the app’s perceived usefulness. The lower scoring and higher variation occur in the first statement defined to assess the acceptance of the adherence strategy, and the lower result is motivated by the need for clearer description of the implemented team competition and scoring strategy, which was deemed too complex for some users. Variability is explained by some participants providing a score of 1, penalizing the app for their experienced frustration while carrying out Task 5. It is also believed that some participants might prefer competing on an individual level basis rather than competing as a team. Figure 21 and Figure 22 illustrate these results.

### 4.6. Qualitative Results Summary

Qualitative data allow the problems and errors that users had during the usability evaluation to be better understood. Although we worked with a heterogeneous profile of participants, there were several aspects that were repeated among the participants’ comments (both written and verbalized).

The main comments were in line with previously described quantitative results, considering the tool as easy to use in overall and perceived as useful. The most repeated suggestion for improvement was related to clarifying and simplifying the gaming approach’s scoring strategy description (both for points and laps). This issue has already been identified within the Task 5 quantitative analysis. Another concept that was mentioned repeatedly for improvement has been related to the challenges functionality location in the application (some participants did not understand it within the follow-up concept, some first looked at the assigned diet section and some even suggested having challenges and other sections summary on the initial dashboard). One user also stated that he would be willing to fill out the diet follow-up weekly or more, but not daily as it was considered quite lengthy.

Next, looking at the participants errors during the tests, some interesting insights have been identified. Errors have been mainly motivated by the following reasons: (i) Incomplete description of questions (e.g., question about supplements or immune system-related illness, where examples were missed by participants); (ii) Questionnaire elements ordering or response options coherence (e.g., vegetable drinks were thought that should go together with milk; no clear differentiation of similar physical activity level questions, asked in consecutive new screens; missed no or 0 response options in some questions or choosing multiple responses, while these were available for other questions; doubts regarding some response should fit/missed further option to provide greater detail for a response); (iii) Interaction-related options (i.e., some horizontal selectors worked across the row, while others only worked if selection was made over the text; some application views were accessed by touching a figure, which was not clear; configurable frequency input was moving too fast for some participants; the decimal separator comma did not work, only period sign functioned appropriately, while the comma is mainly used in Spain); and (iv) Better labelling of menu items and action buttons (e.g., Nutraceuticals was not understood as the nutritional supplements section; some actions for renaming were suggested by participants throughout the tests).

## 5. Discussion

The user evaluation session involved a variety of users from different backgrounds carrying out tasks associated with the obesity self-management approach developed during the OBINTER project. User evaluation was carefully defined to evaluate both the OBINTER approach and the developed solution’s usability, by departing from a well-tested usability protocol [44], and defining representative tasks tackling the OBINTER approach’s different dimensions (see Table 2) and extending the usability questionnaires with approach-specific questions.

Both quantitative and qualitative findings showed that the developed concept and tool was found easy to use and useful, despite the fact that many improvement areas were identified during the formal testing of this first version. In general, task completion times were satisfactory and most users found tasks easier than initially expected as shown by the Single Ease Questions (SEQ). Improvement margin on specific identified usability issues is supported by the analysis of task completion, SEQ and the extended Usability Scale (SUS) results. The need for further maturing the application can be identified on the first SUS question’s lowest score (“I think that I would like to use this system frequently.” mean score 3.8 ± 1) and Net Promoting Score being slightly below (+42) the highest-performing organizations (+50 to +80). However, this conclusion needs to be read with care, as the participants were not explicitly selected to be overweight, obese or willing to start/follow a diet. For this reason, the use or recommendation of this obesity management tool scoring needs to be handled with care.

The qualitative results confirmed and helped to better understand the quantitative results. This was mainly possible thanks to the Think-Aloud Protocol approach adopted for this test. Additionally, the analysis of the qualitative feedback has enabled the identification of issues and improvement areas that arose to multiple participants.

Among the many improvement opportunities identified, the two main issues were related to providing a clearer description for the gaming strategy (it was valued well once the moderator explained it) and the challenges section location. Additionally, some recurrent description, naming and interaction refinement areas were identified. Some of these recommendations may be managed by the technology developers’ side (e.g., interaction means and action names improvement and coherence; making functionality more discoverable and clearer), but some others will require discussion with the nutritional experts team, as there were questions and recommendations specifically related to the nutritional questionnaires and the proposed diet.

## 6. Conclusions and Future Work

In this paper, a new approach for the self-management of chronic obesity through a user-centred digital health solution has been presented. Previous state-of-the-art tools and approaches apply only to the identified techniques (patient-reported outcomes, personalized dietetic and nutritional recommendations incorporating the molecular profile of the individuals, diet adherence control, physical activity tracking, weight evolution control, and adherence strategy) individually or in limited combination, bringing out the deficiencies summarized in Table 1. The main goal the OBINTER approach has is to provide a holistic solution which overcomes those deficiencies while delivering high-grade usability experience.

OBINTER app is proposed as a complement tool for blended care of obese patients. A specific contribution of this tool is the use of scientifically validated algorithms for personalized nutritional recommendations based on metabolic phenotyping of patients to provide the most precise nutritional recommendations to patients. OBINTER app uses molecular data that describe well the subject’s metabolic status (i.e., cell membrane lipidomic data) and represent an overview of the dietary habits followed by an individual during the last four months, and it can characterize metabolomic disorder as well the lack of nutrients in a diet. These bio-indicators, compared with genomics, provide a real time vision of the metabolism change to provide users with a more personalized and effective intervention.

Additionally, since the OBINTER app can make use of wearable devices capable of objectively monitoring physical activities, this can be considered as a step towards the inclusion of this type of person-generated data (with the required robustness and quality) within clinical care processes or as part of research on the origins and progression of a disease.

To validate the OBINTER approach, a formally defined usability evaluation has been carried out. From the results of the user evaluation, it can be concluded that the digital approach proposed by OBINTER for obesity self-management has been found easy to use and useful, while it has allowed to identify a rich set of improvements for its future versions. Usability testing is an important step towards refining the development of the OBINTER app, which can be used in the self-management of obesity. To get a further understanding of the developed approach, we plan to extend usability evaluation questionnaires with questions addressing identified main usability issues (i.e., providing a clearer description for the gaming strategy and the challenges section location), with the aim of getting more insight regarding them. Additionally, the inclusion of A/B testing [56] with alternative solutions for the identified usability limitations will be arranged to understand user preferences.

From the clinical point view, a clinical trial was planned, supported by the presented mobile application, to be conducted during year 2020, but the COVID-19 pandemic forced it to be postponed until the end of year 2020. This study will involve 100 individuals randomly chosen from the Basque Country and will be provided with simple mechanisms during the experiment duration, to allow them to yield usability feedback at any time. Additionally, the app has been prepared to capture and evaluate user’s adherence to the app during the clinical trial or production use automatically.

Based on the obtained user evaluation results, the planned future work will start targeting the detected usability issues and continue to improve the platform. On the app side, a clearer use instruction will be provided to the user (evaluating first time use aid as guided tour), and a possible relocation of sections will be carried out. Furthermore, the provided nutritional content will be revised, both the questionnaires and the recommendations. Additionally, we plan on expanding the use of the tracker’s data by increasing the scope of the data that are used in the OBINTER platform and developing a web platform to visualize and explore the data of each patient.

From a research and technical innovation perspective, the main identified goal is to investigate on machine learning algorithms to further automate tracking and adherence control dimensions. Diet adherence control through automatic food analysis technology [57] and the generation of more personalized challenges will be especially targeted.

## Figures and Tables

**Figure 1 sensors-20-05060-f001:**
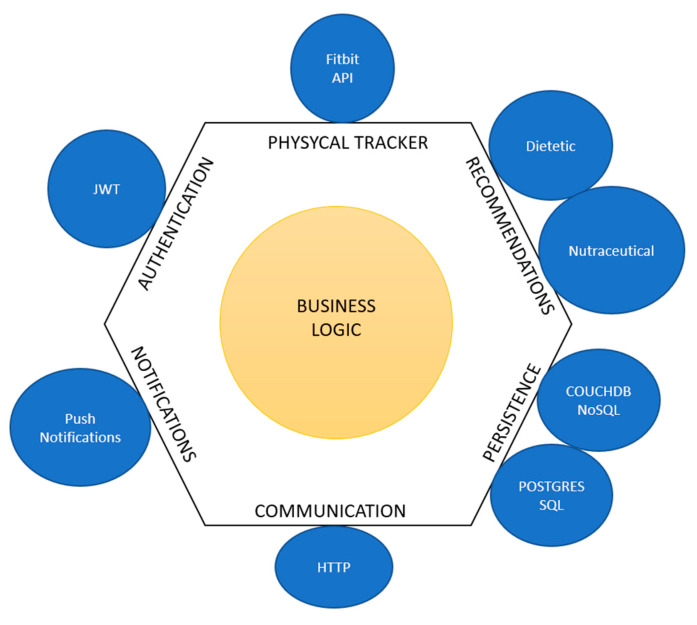
Adaptation of Cockburn’s hexagonal architecture to OBINTER approach.

**Figure 2 sensors-20-05060-f002:**
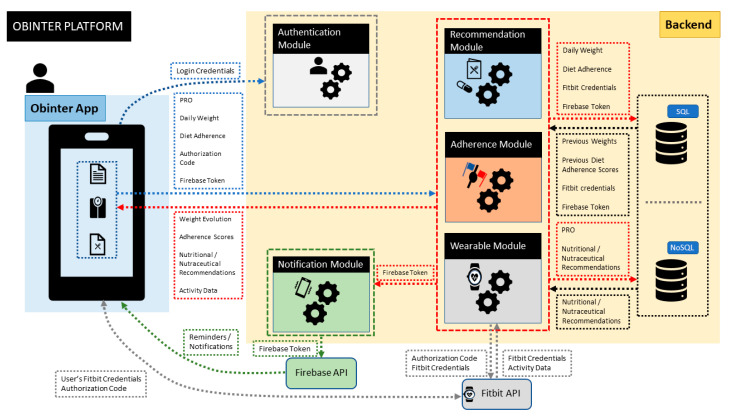
High-level abstraction of OBINTER platform architecture. The OBINTER platform is composed of two main physical blocks (i.e., Obinter App and Backend) together with the Firebase API (for notifications) and Fitbit API (for wearable data integration). Next, the different modules depicted within the backend represent the logical functionalities implemented by the OBINTER platform.

**Figure 3 sensors-20-05060-f003:**
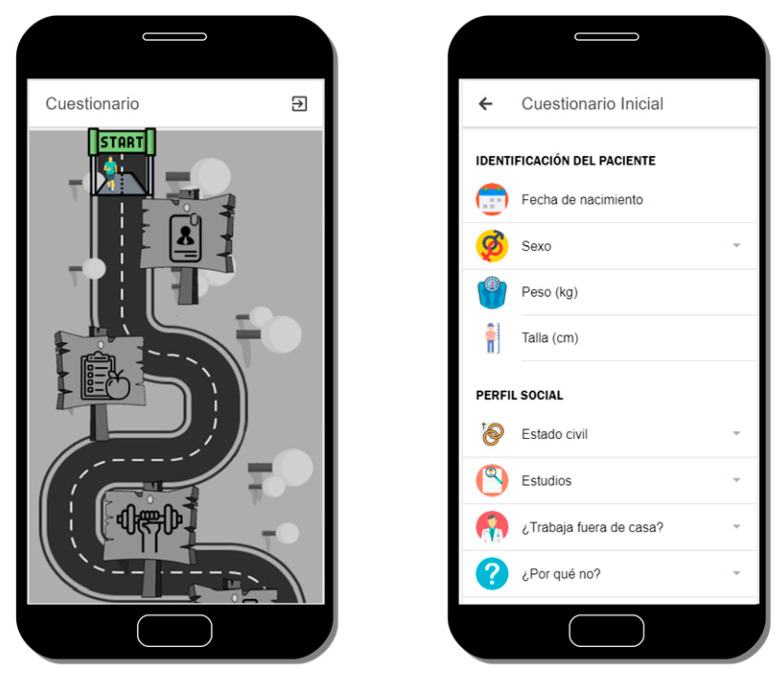
Main interface to navigate through all the life habits questionnaire’s sections (left) and detail of the first section of the questionnaire (right).

**Figure 4 sensors-20-05060-f004:**
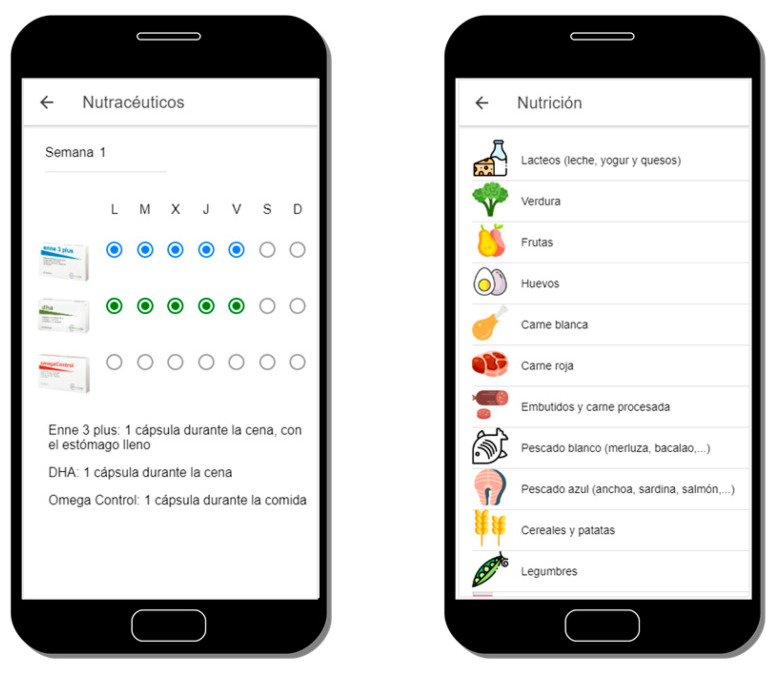
View of a nutritional supplement (nutraceutical) recommendation (left) and the main screen of the dietetic recommendation (right).

**Figure 5 sensors-20-05060-f005:**
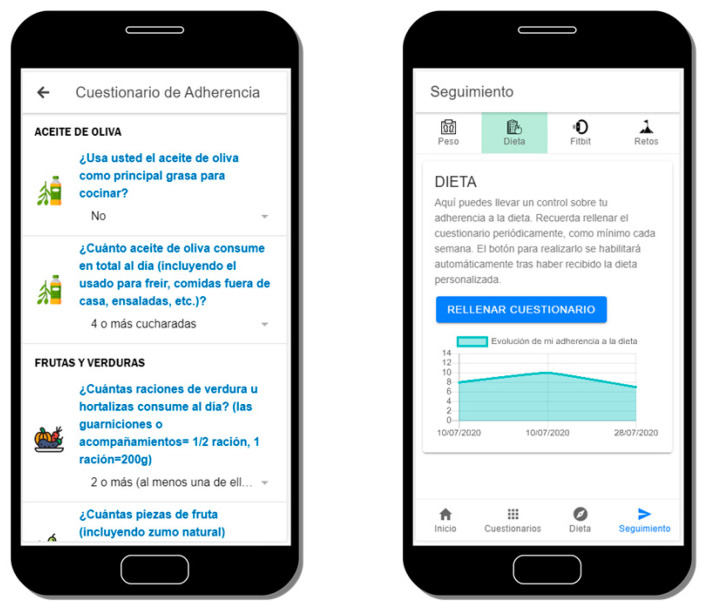
Detail of the Mediterranean Diet adherence questionnaire (left) and its corresponding score evolution (right).

**Figure 6 sensors-20-05060-f006:**
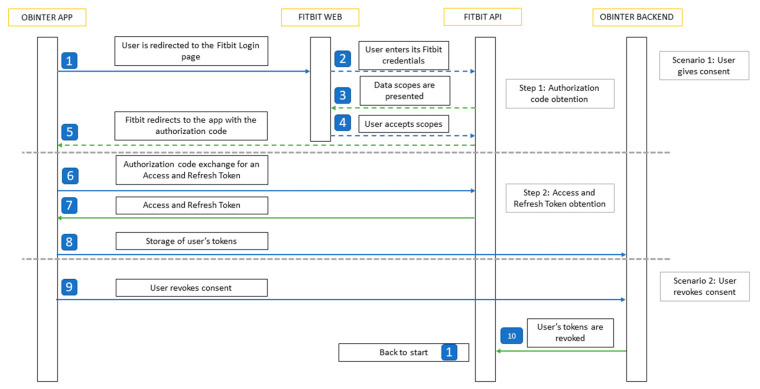
Sequence diagram describing the workflow to integrate Fitbit’s user authorization rotocol.

**Figure 7 sensors-20-05060-f007:**
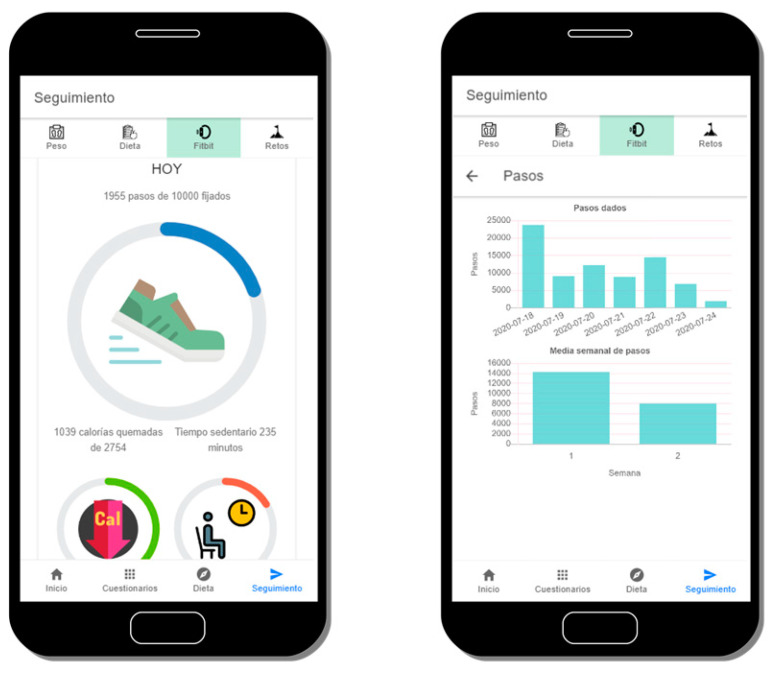
View of the current Fitbit’s data (left) and detailed view of the daily steps aggregated data (right).

**Figure 8 sensors-20-05060-f008:**
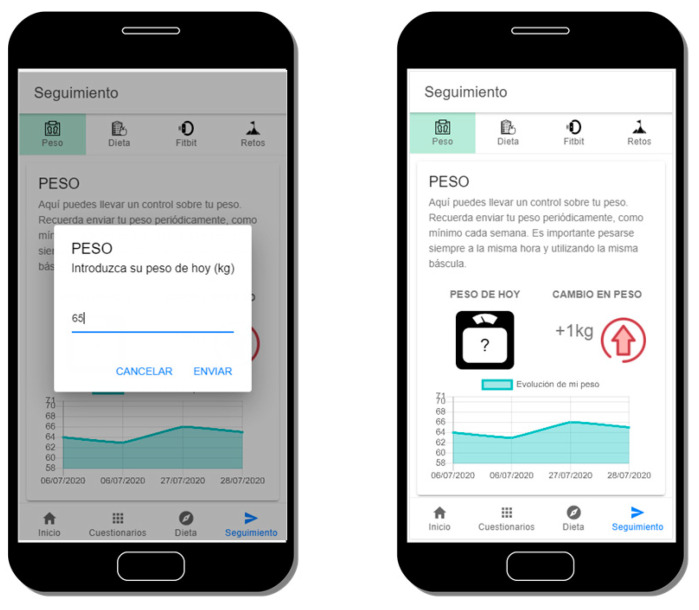
Input to insert users’ weight (left) and evolution weight (right).

**Figure 9 sensors-20-05060-f009:**
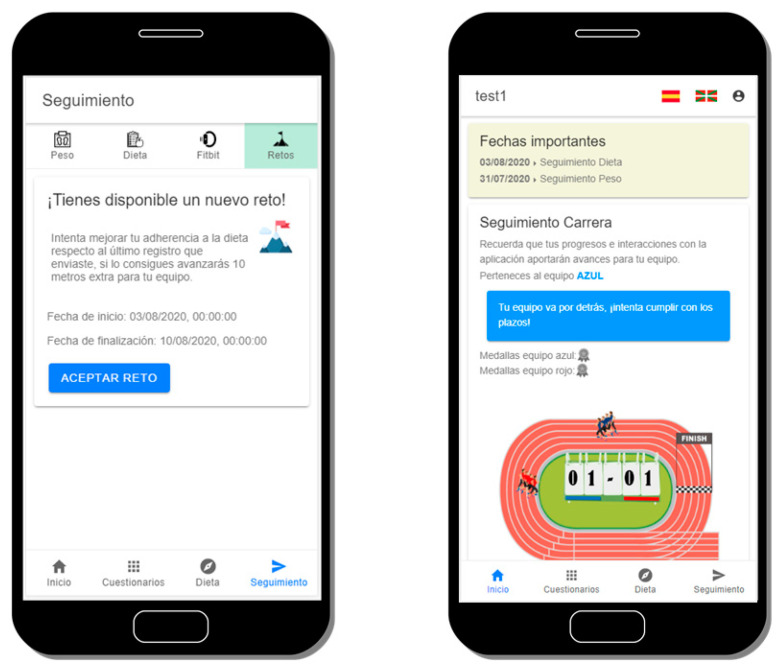
Capture of the challenge delivery system (left) and overview of the virtual race game (right).

**Figure 10 sensors-20-05060-f010:**
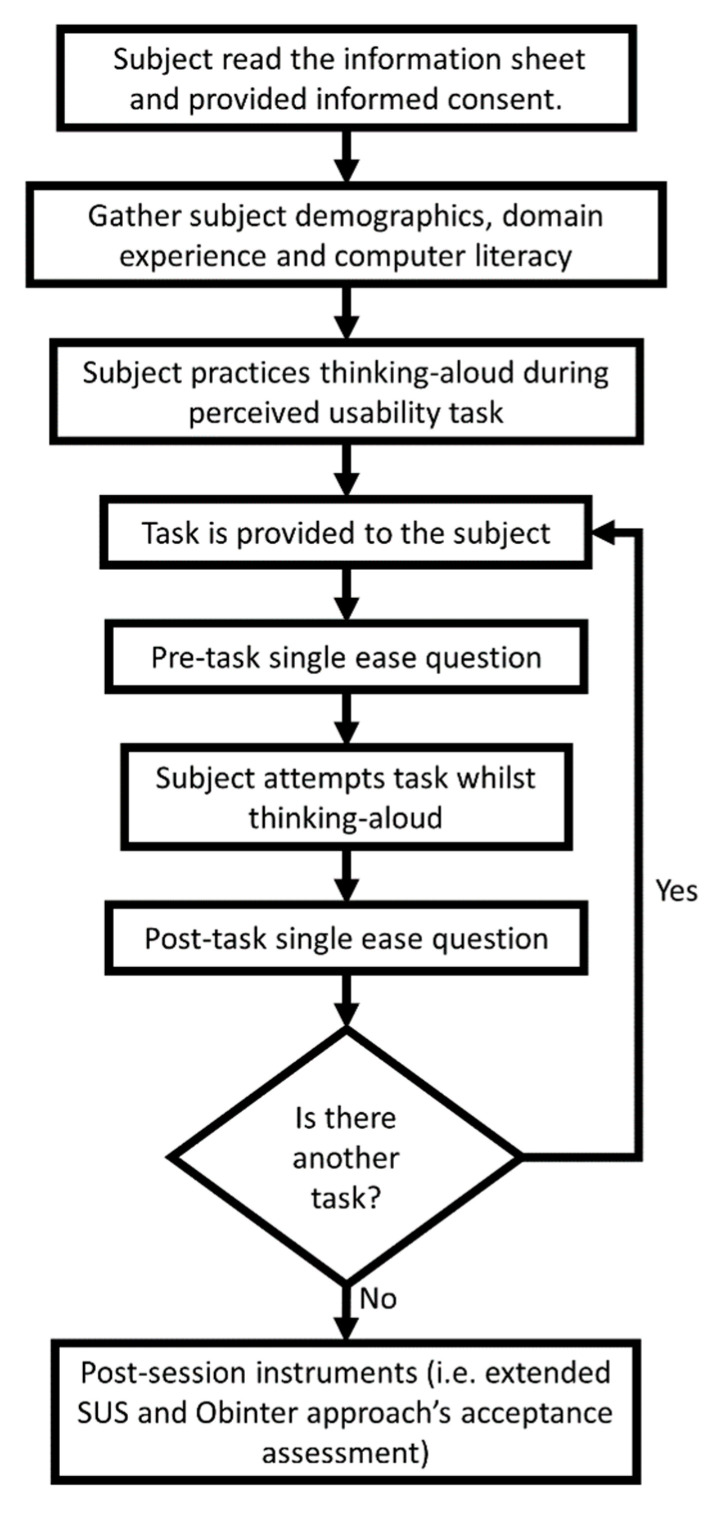
Defined Usability testing protocol diagram. This figure has been adapted from [44].

**Figure 11 sensors-20-05060-f011:**
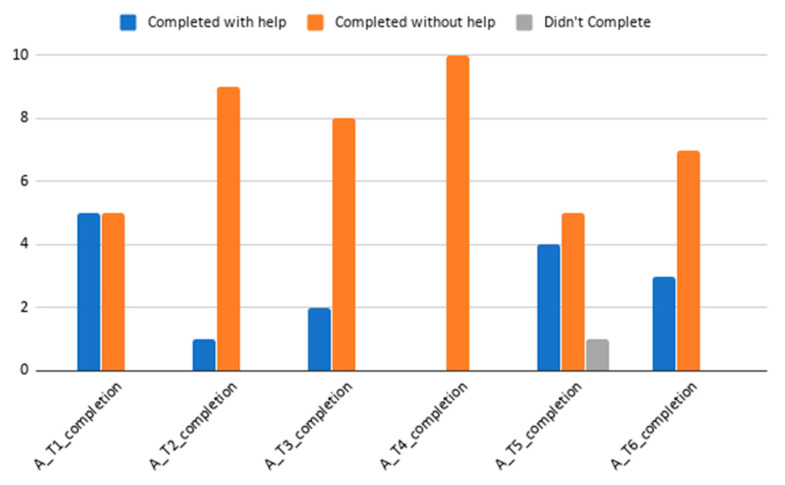
Task completion results. Bar chart plot showing task completion results and if help was required for task completion.

**Figure 12 sensors-20-05060-f012:**
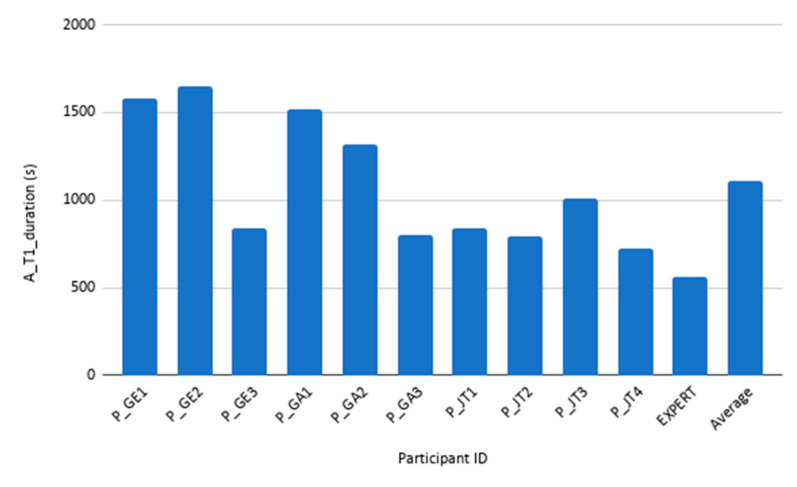
Task 1 duration results. Different participant’s duration for Task 1, together with expert’s and average duration.

**Figure 13 sensors-20-05060-f013:**
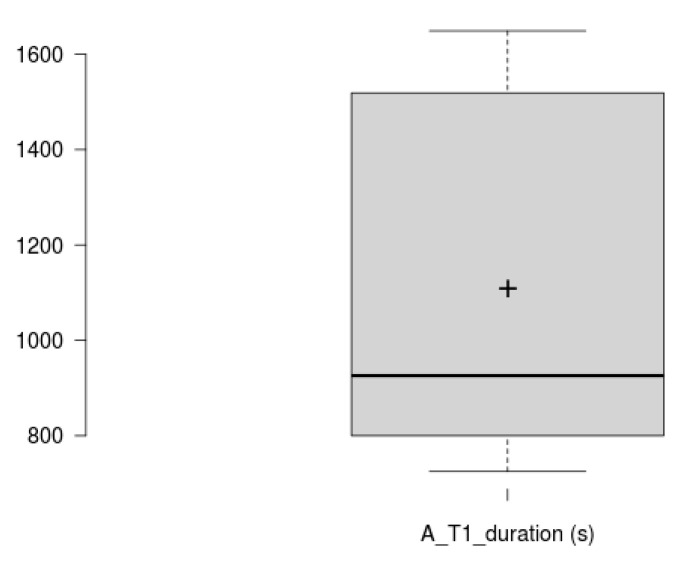
Task 1 duration variability. Box plot showing overall duration scores for Task 1. This does not, however, include expert user’s duration. The plus sign denotes the average value.

**Figure 14 sensors-20-05060-f014:**
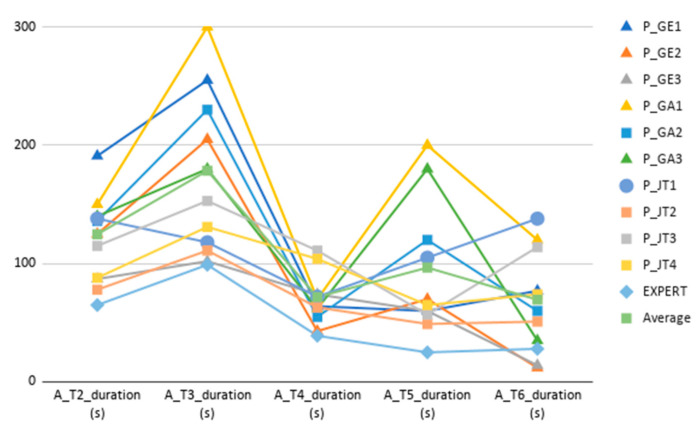
Tasks 2–6 duration results. Different participants’ durations for tasks 2–6, together with the expert’s and average durations.

**Figure 15 sensors-20-05060-f015:**
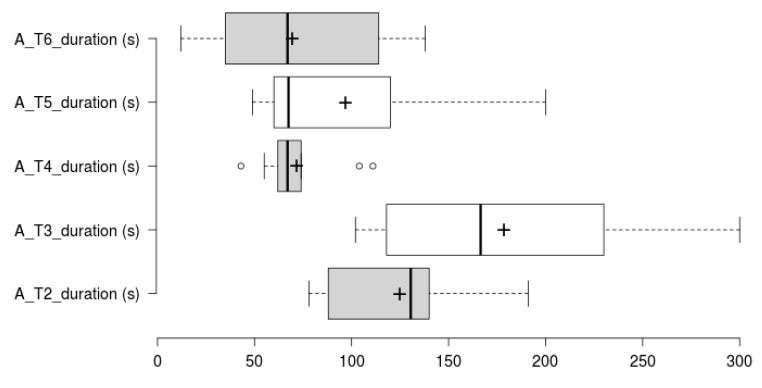
Tasks 2–6 duration variability. Box plot showing overall duration scores for task 2–6. The plus sign denotes the average value. It does not include expert user’s duration.

**Figure 16 sensors-20-05060-f016:**
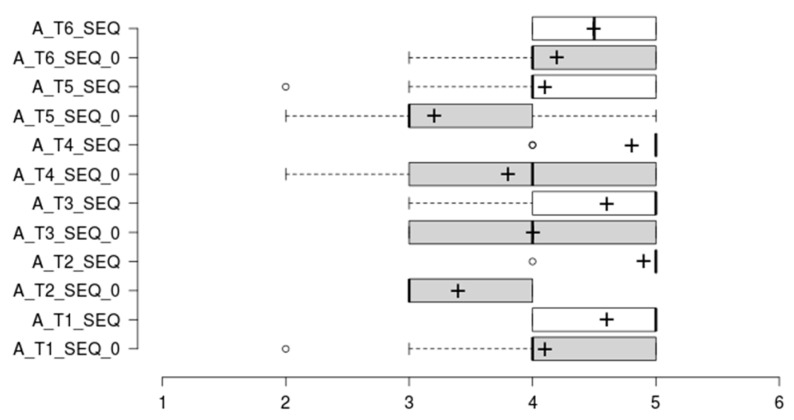
Task Difficulty Ratings. Box plots comparing the anticipated task difficulty ratings (SEQ_0) and the post-task difficulty ratings (SEQ), where 1 = ’Very Difficult’ and 5 = ’Very Easy’. The plus sign denotes the average value.

**Figure 17 sensors-20-05060-f017:**
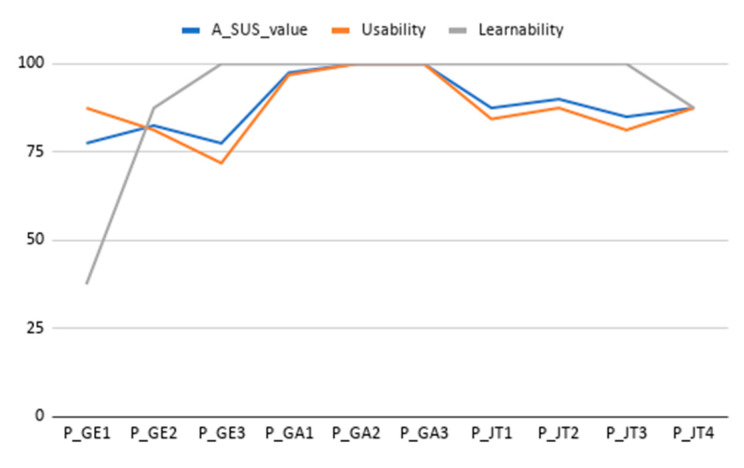
Results from the SUS scores per participant (Overall Score, Usability Dimension, Learnability Dimension). Line chart shows the score achieved in each dimension for each participant and dimension. Value range is from 0 (minimum score) to 100 (maximum score).

**Figure 18 sensors-20-05060-f018:**
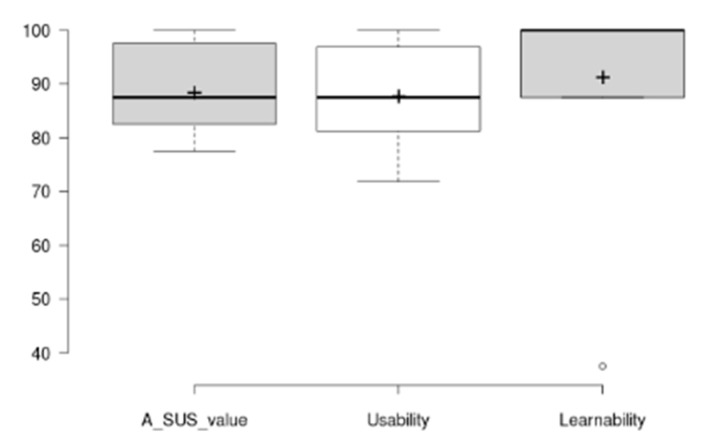
SUS results. Box plot showing overall scores and variability for the different SUS dimensions all tasks). Th plus sign denotes the average value.

**Figure 19 sensors-20-05060-f019:**
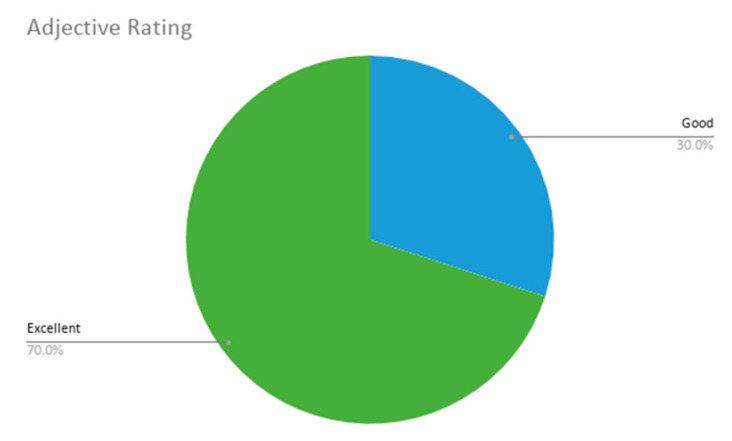
Adjective rating results. The pie chart with adjective ratings (possible options in improving value order: Worst Imaginable, Awful, Poor, OK, Good, Excellent, Best Imaginable); from the possible options, only Good and Excellent were used.

**Figure 20 sensors-20-05060-f020:**
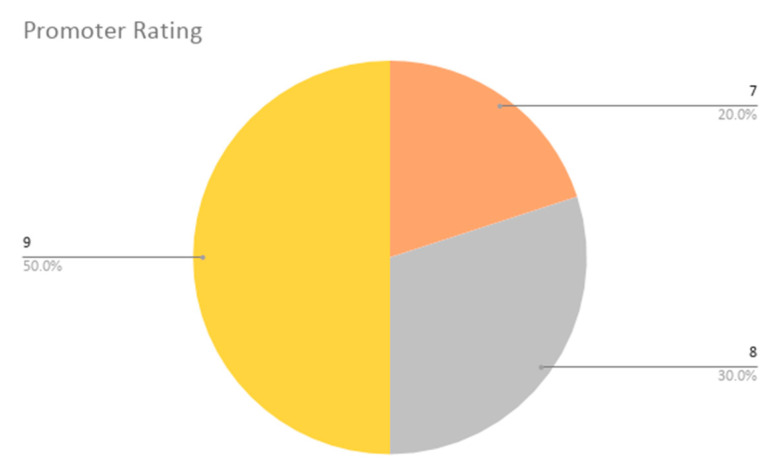
Promoting rating results. The pie chart with promoter rating scores, where 1 = ’Not at all likely Disagree’, 5 = ’Neutral’ and 10 = ’Extremely likely’. From the possible options, only 7, 8 and 9 values were given.

**Figure 21 sensors-20-05060-f021:**
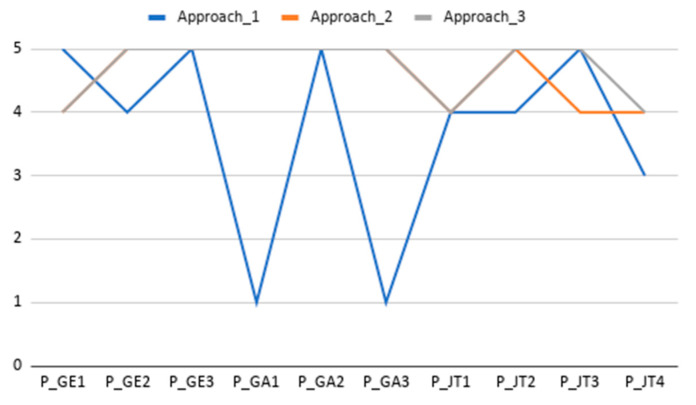
OBINTER Approach-related specific statement assessment results per participant. Line chart shows the score achieved in each question for each participant, where 1 = ‘Strongly Disagree’ and 5 = ‘Strongly Agree’.

**Figure 22 sensors-20-05060-f022:**
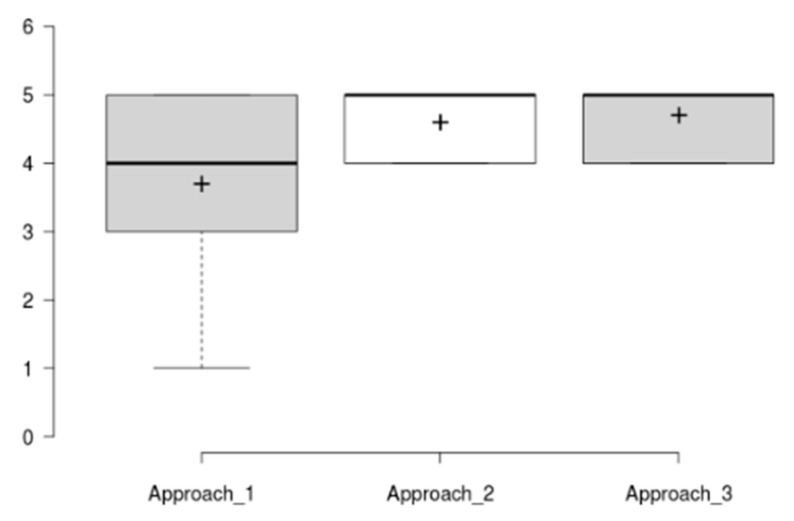
OBINTER Approach-related specific statement assessment results. Box plot showing overall scores and variability for the different results. The plus sign denotes the average value.

**Table 1 sensors-20-05060-t001:** Comparative of the different approaches from the state-of-the-art in relation with the OBINTER app.

Approach	PRO	Personalized Dietetic/w/Metabolic Data	Diet Adherence Control	Weight Control	PA tracking	Push Notifi-cation	Adherence/Gaming Strategy
From [16] Livestrong	No	Not from experts/No	Yes	Yes	Not device	Yes	Yes
From [16] myWW app	No	Not from experts/No	No	Yes	Sync device	Yes	No
From [18] SMART MOVE	No	No/No	No	No	Not device	No	Yes
From [21] Digital Health Platform	Yes	Yes/Yes	No	Yes	Not device	No	No
From [23] Lose It!	No	Not from experts/No	Yes	Yes	Sync device	Yes	No
From [23] MyNetDiary	No	Not from experts/No	Yes	Yes	Sync device	Yes	No
From [23] MyFitnessPal	No	Not from experts/No	Yes	Yes	Sync device	Yes	No
OBINTER App	Yes	Yes/Yes	Yes	Yes	Sync device	Yes	Yes

**Table 2 sensors-20-05060-t002:** Tasks tested by each person, defined within the usability testing protocol.

Approach Dimensions	Task No.	Task Description
Patient-Reported Outcome	1	Fill initial questionnaire on patient’s background
Personalized Dietetic Recommendations, Personalized Nutraceutical Planning	2	Check background-based assigned diet and nutraceutical planning
Weight evolution Control, Diet Adherence Control	3	Fill weight and diet follow-up information
Physical activity tracking	4	Check steps number, burned calories and sedentary time
Adherence Strategy	5	Check assigned team and how well you are doing (adherence)
Adherence Strategy	6	Check, read and accept a proposed wellbeing challenge

**Table 3 sensors-20-05060-t003:** Demographics of participants involved in the user evaluation of the OBINTER App (*n* = 10).

Demographic	Result
Gender	3 males, 7 females
Age	Mean 36.9 ± 12.02 years of age
Occupation	2 Administrative Assistants, 3 Researchers, 1 Teacher, 2 Workers, 1 Insurance Expert, 1 Student
Primary Expertise	1 Human Resources/1 Economics/3 Biomedicine/1 Languages/2 Manufacturing/1 Automotive/1 Computer Vision
First language	4 Basque, 6 Spanish
Average computer usage per week	Mean 32.5 ± 14.64
Average Internet usage per week	Mean 48.1 ± 31.67
Average level of computer literacy	Mean 3.5 ± 1.08
	Mode: 4
Familiarity with diet/wellbeing tracking apps	Mean 1.8 ± 1.13
	Mode: 1

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
