# Peer review of "OBINTER: A Holistic Approach to Catalyse the Self-Management of Chronic Obesity"

_sensors, 2020, doi:10.3390/s20185060_

Round 1
Reviewer 1 Report
The manuscript has merit and presents an interesting solution particularly useful to monitor obesity and promote good practices in health and well being, possibly having a positive impact on the health status of the population and on the economy of the health systems.
Overall, I recognize just three possible drawbacks in the manuscript, that in my opinion should be addressed:
- Nowadays, there are hundreds of Apps dealing with the health and well-being promotion in nearly every field, also in obesity-related questions. Please, point out, even schematically, and after generating a discussion, the main points that differentiate the present App with respect to other, existing solutions, and eventually stress the concept on how this App “improves” existing functionalities and/or “fill in some existing gaps”.
- Albeit promising, the App was yet tested on a very limited group of persons. The collection of more data would help in better understanding strengths, weaknesses and usability limitations of this tools. Please, take into account and eventually discuss.
- Please, hypothesize how this App can be improved in future developments (i.e., embedding consumer technologies solutions to objectivize the data collected – see, for just one example, Aslan et al., IEEE ICCE-Berlin 2018.
Finally, English language and grammar should be carefully revised. Some typos are also present throughout the text. Please, check.
Reviewer 2 Report
This article designs a monitoring platform of adherence strategies to reduce the risks of Obesity. The authors design and validate a holistic approach to monitor patients. I think that this platform is well designed and described. However, I recommend to do the following issues:
1) The state of the arts is not well done. For example, existing solutions are not well described and discussed to show the new values added by this article. I propose to add a new section to detail the state of the arts.
2) In addition, a section of mobile applications can be added to clarify the existing solutions.
3) The authors proposed a holistic approach, but the comparison with existing approaches is not done. Therefore, it is not clear how authors can argue their approach.
4) Organization of paper can be added and detailed in introduction part.
5) it is not clear how the authors get the permission to do the experimentation to validate the platform. I am wondering how the authors get this permission.
Round 2
Reviewer 1 Report
My concerns were successfully answered. Congratulations for your careful revision of this paper.